# Fine-scale modelling finds that breeding site fragmentation can reduce mosquito population persistence

Clare P. McCormack [1], Azra C. Ghani[1] & Neil M. Ferguson [1]

Fine-scale geographic variation in the transmission intensity of mosquito-borne diseases is primarily caused by variation in the density of female adult mosquitoes. Therefore, an understanding of fine-scale mosquito population dynamics is critical to understanding spatial heterogeneity in disease transmission and persistence at those scales. However, mathematical models of dengue and malaria transmission, which consider the dynamics of mosquito larvae, generally do not account for the fragmented structure of larval breeding sites. Here, we develop a stochastic metapopulation model of mosquito population dynamics and explore the impact of accounting for breeding site fragmentation when modelling fine-scale mosquito population dynamics. We find that, when mosquito population densities are low, fragmentation can lead to a reduction in population size, with population persistence dependent on mosquito dispersal and features of the underlying landscape. We conclude that using non-spatial models to represent fine-scale mosquito population dynamics may substantially underestimate the stochastic volatility of those populations.

[1] MRC Centre for Global Infectious Disease Analysis, Department of Infectious Disease Epidemiology, Imperial College London, London W2 1PG, UK. Correspondence and requests for materials should be addressed to C.P.M. (email: c.mccormack14@imperial.ac.uk)

Mosquito-borne diseases are a major public health challenge in many countries worldwide. The year 2017 saw an estimated 219 million cases of malaria globally[1] and the World Health Organization currently estimates that half of the world's population is a risk from infection by dengue, with the virus now endemic in more than 100 countries[2]. The transmission intensity of both dengue and malaria varies geographically, with spatial heterogeneity evident at broad and fine spatial scales[3–5]. Although a wide range of environmental and social factors including climate[6,7], human mobility[8], and levels of urbanization[9,10] play an important role in determining transmission intensity, the primary cause of spatial heterogeneity in transmission intensity is variation in the density of female adult mosquitoes[11,12]. Thus, critical to an understanding of the drivers of spatial heterogeneity in disease transmission and persistence at fine spatial scales, is an understanding of the fine-scale dynamics of mosquito populations[13,14].

Adult mosquito density in any given environment is largely determined by two key factors. First, the mortality rate of adult mosquitoes and, second, the rate at which new adult mosquitoes emerge from larval breeding sites. The capacity of larval breeding sites to produce new adult mosquitoes is constrained however by the availability of resources at those sites. Such resources are inherently limited, so any one larval breeding site has a limit — termed the carrying capacity — to the size of the larval population it can support. Thus, larval populations are constrained and it is generally assumed that such regulation is dominated by density-dependent intraspecific competition, whereby larvae in a single breeding site compete for food and other resources[15–19]. Although this has the primary effect of increasing larval mortality, it may also lead to longer development times and a reduction in the size of adult mosquitoes[15,16,18,20,21].

Characteristics of desired breeding sites vary with mosquito species. Although Anopheles mosquitoes, which transmit malaria, typically breed outdoors in naturally occurring freshwater habitats such as pools, ponds and marshes[22–24], Aedes aegypti, the primary vector of dengue, are adapted to an urban environment. Thus, they primarily breed in rain-filled, man-made habitats such as household containers and discarded tyres[15,25,26], and there can be several breeding sites per household. Larval breeding sites for both species are therefore often highly fragmented[15,22,23,25], with mixing among the population determined by the dispersal[27,28] and oviposition behaviour[29,30] of the mosquito.

This fragmented structure leads to both temporal and spatial variation in the availability, size and quality of larval breeding sites across any given landscape[15,22,23,25]. Empirical research has found that A. aegypti exhibit considerable spatial variation in abundance, even at the individual household level, with clustering among households of similar levels of carrying capacity often occurring[25,26,31–34]. Furthermore, recent analysis of the spread of the bacterium Wolbachia among local A. aegypti populations in northern Australia has shown that both the pattern and speed of spatial spread of Wolbachia is highly heterogeneous, largely owing to environmental factors, including fine-scale variation in A. aegypti population density[35]. Empirical evidence suggests that abundance of Anopheles mosquitoes is also spatially heterogeneous at fine spatial scales, owing to the availability and productivity of larval habitats, and proximity to human settlements[23,36–38].

Whether and how the underlying fragmented structure of larval populations is accounted for in mathematical models of mosquito population dynamics is governed by assumptions made with respect to the spatial structure of larval breeding sites and how the dynamics of density-dependent competition are represented. Owing to the fragmented structure of larval breeding sites, the intensity of density-dependent competition will vary across a landscape, with more densely populated habitats subject to a greater degree of competition. However, currently, mathematical models of dengue and malaria transmission, which consider larval population dynamics[39–43], tend to adopt a very simple representation of density-dependent competition. Namely, the entire larval population is treated as a well-mixed population coming from a single large breeding site and the larval mortality rate is assumed to increase linearly with larval population size. In reality, however, the dynamics of density-dependent competition across a fragmented landscape are likely to be much more complex. Moreover, density-dependent competition is a nonlinear process, and hence it cannot be assumed that modelling a single large well-mixed larval population will generate the same dynamics as a model which explicitly represents a network of fragmented local populations.

Ecological research has shown that habitat fragmentation can lead to increased population instability and decreased population persistence, as a combination of demographic and environmental stochasticity places small local populations at continual risk of extinction[44–47]. If similar results hold for mosquito populations, this could have important implications for mathematical models of dengue and malaria, which consider larval population dynamics, particularly in the context of estimating vector population persistence at fine spatial scales. Hence, to explore the impact of accounting for the fragmented structure of larval populations when modelling the fine-scale dynamics of mosquito populations, we developed a stochastic metapopulation model of mosquito population dynamics and considered the effects of habitat fragmentation on the dynamics observed by modelling the same mosquito population at different levels of spatial granularity, corresponding to different levels of fidelity in representing the true underlying spatial structure of mosquito populations. We explored how habitat fragmentation, features of the underlying landscape and the level of spatial granularity affect the dynamics observed, with the overall aim of understanding the role and importance of spatial structure in shaping the dynamics of mosquito populations at fine spatial scales. We find that failing to account for larval habitat fragmentation may substantially underestimate the vulnerability of local mosquito populations to extinction and conclude that how spatial structure is represented in our model strongly influences our understanding of fine-scale mosquito population dynamics.

## Results

**Impact of fragmentation.** We first considered the dynamics of an established mosquito population in a homogeneous landscape. We observed that fragmentation may lead to a reduction in both total population size and patch occupancy levels, compared with the single-patch model (Fig. 1a, b). The magnitude of this effect depended on the carrying capacity of each patch (represented as the deterministic equilibrium larval population size in Fig. 1). For the non-spatial single-patch model, population size scaled linearly with carrying capacity as expected, whereas the relationship was nonlinear for the metapopulation model, especially in the absence of dispersal. As expected, the largest differences occurred when the carrying capacity of individual patches in the metapopulation was low, as the probability of stochastic extinction is then highest.

Adult mosquito dispersal counteracted these effects by enabling extinct local populations to be reseeded, thereby rescuing these populations and increasing population persistence and patch occupancy. For patches with low carrying capacity and hence an unstable local population, this recurring cycle of extinction and recolonization leads to a continuous fluctuation in patch occupancy at the local level. However, at the global level, patch occupancy levels remained stable, indicating that a balance

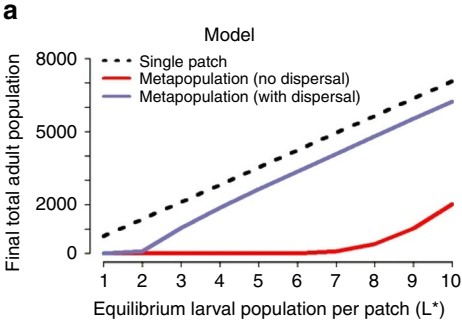
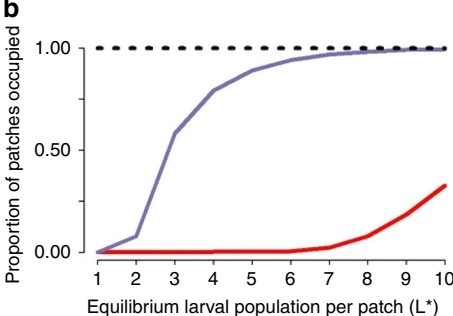

**Fig. 1** Impact of fragmentation on an established mosquito population. Mean final total population size and patch occupancy for a mosquito population modelled as a single patch and as a metapopulation on a 32 × 32 grid, using a dispersal rate of 0.08 per day and a mean dispersal distance of 5 patches. The initial population of each patch is at its deterministic equilibrium at the start of each simulation and the total initial adult population in the single patch model is 32 × 32 times the initial adult population of each patch in the metapopulation model. The dashed black line represent the results observed under the single patch model. Figures **a**, **b** compare the results observed for a homogeneous landscape with no temporal or spatial variation in larval-carrying capacity. For each scenario, the mean was calculated across 1000 realizations of the stochastic model

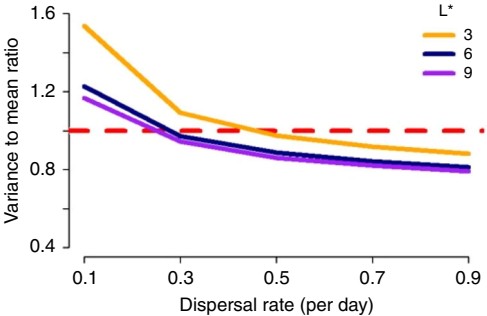

**Fig. 2** Impact of increasing the dispersal rate. Variance to mean ratio of a local larval population at the final time step in a metapopulation comprised 1024 (32 × 32) patches with no temporal or spatial heterogeneity in carrying capacity, and where each patch has an equilibrium larval population of 3, 6 or 9 larval mosquitoes. The dashed red line denotes the crossover point from over- to under-dispersion relative to the Poisson distribution. For each scenario, the mean and variance were calculated across 1000 realizations of the stochastic model

between extinction and recolonization across the metapopulation is maintained, allowing the overall population to persist. Further stabilizing mechanisms of the metapopulation were evident from exploring the relationship between the degree of dispersal and the variability of local larval populations. Indeed, dispersion can make individual patch populations more stable than isolated patches. Examining the variance to mean ratio of the larval population in individual patches, we observed a move from over-dispersion to under-dispersion as the rate of dispersal increased (Fig. 2).

Next, we examined the effect of landscape fragmentation on population invasion, while accounting for seasonal variation in carrying capacity. In the single patch model, we observed fast population growth, with almost all model runs resulting in population persistence (Fig. 3). However, in a fragmented landscape, a different picture emerges, with fragmentation hindering population persistence and growth, particularly when the dispersal rate is low and movement beyond the seeded patch is limited. When patches have a low carrying capacity, the risk of extinction from stochastic effects is high, which in turn reduces the likelihood of sustained invasion upon seeding (Fig. 3a) and the speed of population spread across the landscape (Fig. 3b, c). Moreover, for model runs that were successful in achieving population persistence, fragmentation reduced the level of population growth and increased the amount of variability in

the population, prior to the population stabilizing, compared with the single patch model (Fig. 3b–d). Seasonal troughs in carrying capacity posed a larger barrier to spatial spread during the early stages of population growth in a fragmented landscape, further slowing the speed of spread (Figs. 3b and 4). However, the speed of spread across the landscape was highly dependent on mean dispersal length, with an increased range of dispersal across the landscape accelerating the speed of invasion, even at a low rate of dispersal (Fig. 4).

When modelling the dynamics of an established mosquito population or those of a newly seeded population, the dynamics observed were highly dependent on the level of spatial granularity assumed in the model. As spatial granularity is reduced, this increases patch size and thus the carrying capacity of individual patches. This in turn reduces the probability of stochastic extinction in individual patches. Thus, in a context where dispersal rates were low to moderate, a reduction in the level of spatial granularity largely resulted in increases in long-term mean population sizes, population persistence, and patch occupancy, with the behaviour of the metapopulation largely tending towards that of the single-patch model as the level of granularity is further reduced (Fig. 4).

A notable exception to this pattern was observed when examining the effect of spatial granularity on invasion dynamics, as the dynamics observed depend heavily on the level of spatial granularity, and the dispersal rate and kernel, and thus a more complex picture emerges. For both low and high values of single-patch-carrying capacity, if dispersal across the landscape at the highest level of spatial granularity is very local then, as granularity is reduced, movement beyond the seeded patch becomes less likely. Therefore, at lower levels of granularity, population spread and growth beyond the seeded patch may not occur, despite a very low risk of extinction through stochastic effects in neighbouring patches (Fig. 4c, d). Thus, behaviour tending towards the single-patch model as granularity is reduced is not necessarily guaranteed and relies on sufficient movement between patches at lower levels of granularity.

**Landscape heterogeneity**. The impact of non-clustered heterogeneity in carrying capacity across the landscape on the population dynamics observed was dependent both on the mean value of carrying capacity and the underlying dispersal dynamics. Increasing inter-patch variability, while keeping the total carrying capacity fixed, increases the risk of local stochastic extinction as a higher proportion of patches have low carrying capacities. However, this can have either positive or negative effects on

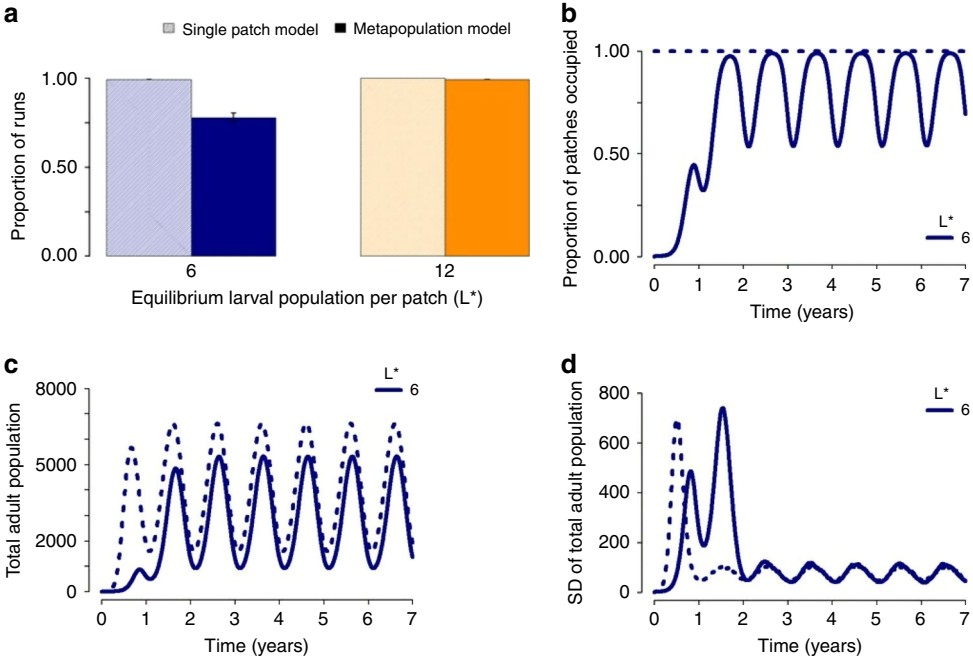

**Fig. 3** Impact of fragmentation when seeding a population. Population dynamics observed when a single patch in a landscape with temporal variation in carrying capacity is seeded with an adult mosquito population, and is modelled as a single patch (dashed lines) or as a metapopulation (with a dispersal rate of 0.08 per day) on a 32 × 32 grid (solid lines). We consider the dynamics for landscapes with a mean equilibrium larval population of 6 or 12 larvae per patch (L*), seeded with 6 or 12 adult mosquitoes, respectively, and where the amplitude and phase of seasonal variation in carrying capacity are 0.7 and 0.5, respectively. The results are compared with respect to **a** the mean proportion of model runs in which population persistence was achieved — i.e., final total adult population size >1 (mean and SD are plotted), **b** the mean proportion of patches occupied, **c** mean total adult mosquito population size, and **d** the SD of total adult mosquito population size. For each scenario, the mean and variance were calculated across 1000 realizations of the stochastic model

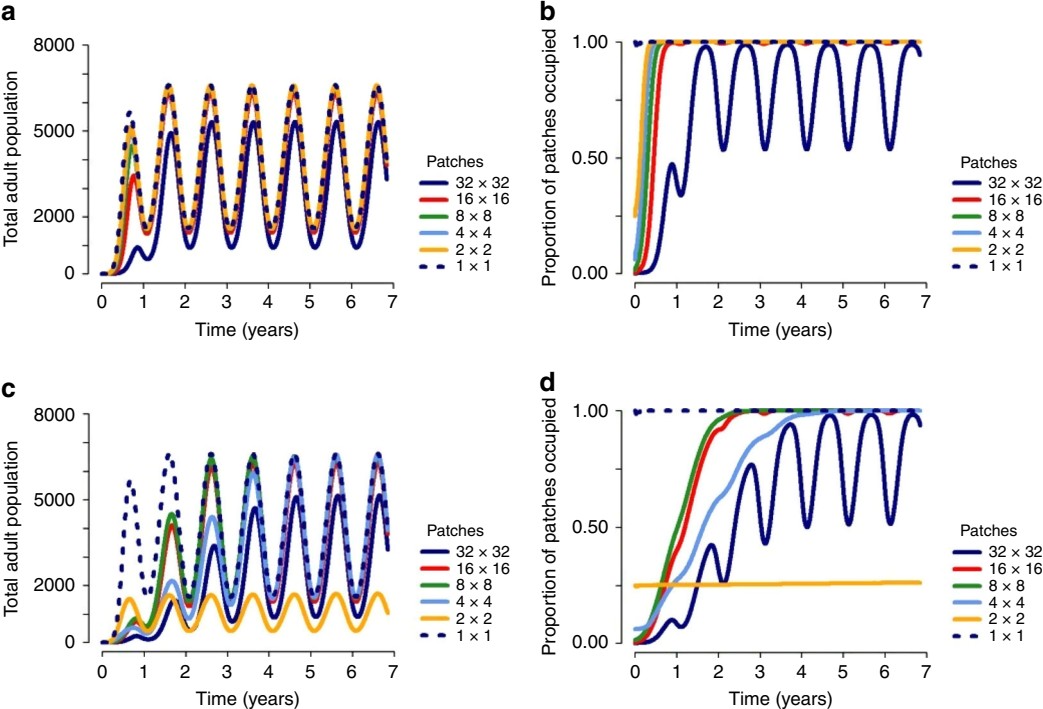

**Fig. 4** Modelling at different levels of spatial granularity. Population dynamics observed when a landscape with temporal variation in carrying capacity and an equilibrium mean larval population of six larvae per patch (at the 32 × 32 level) is seeded with an adult population of six mosquitoes and modelled at different levels of spatial granularity. The dispersal rate was 0.08 per day. **a**, **b** Correspond to dynamics observed when the mean dispersal length (at the 32 × 32 level) is five patches and **c**, **d** correspond to the dynamics observed when the mean dispersal length (at the 32 × 32 level) is one patch. The maximum dispersal length (at the 32 × 32 level) is set to 32 patches. For each scenario, the mean was calculated across 1000 realizations of the stochastic model

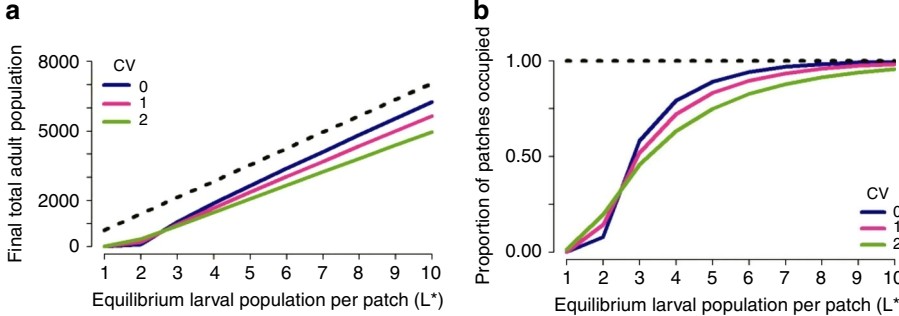

**Fig. 5** Impact of fragmentation on an established mosquito population. Mean final total population size and patch occupancy for a mosquito population modelled as a single patch and as a metapopulation on a 32 × 32 grid, using a dispersal rate of 0.08 per day and a mean dispersal distance of five patches. The initial population of each patch is at its deterministic equilibrium at the start of each simulation and the dashed black line represent the results observed under the single-patch model. Figures **a**, **b** compare the single-patch model to the metapopulation model with dispersal, for a heterogeneous landscape with the level of spatial heterogeneity in carrying capacity across patches characterized by the coefficient of variation (CV) and no temporal variation in carrying capacity. For each scenario, the mean was calculated across 1000 realizations of the stochastic model

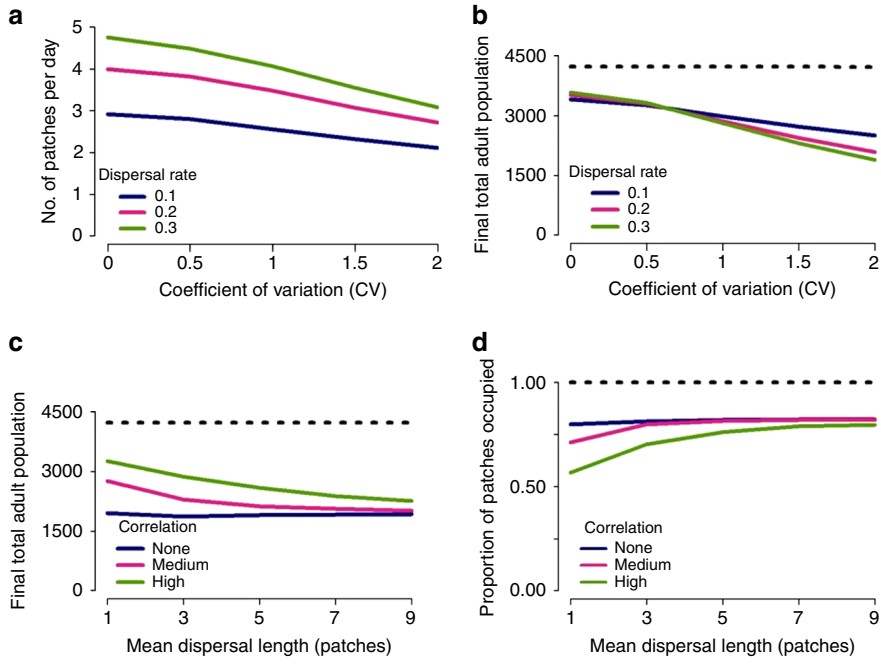

**Fig. 6** Heterogeneous landscapes. Population dynamics observed in a spatially heterogeneous landscape comprised 1024 patches, with a deterministic mean equilibrium larval population of 6 larvae per patch, spatial heterogeneity characterized by the coefficient of variation relative to this mean value and no temporal variation in carrying capacity. **a** Speed of population spread in a landscape seeded with six adult mosquitoes. Speed is characterized in terms of the average number of patches travelled per day, calculated by dividing the time taken until all patches have been occupied at least once by the number of patches in the metapopulation. **b–d** Dynamics of an established mosquito population. For **c**, **d**, a coefficient of variation of 2 and a dispersal rate of 0.3 per day was used. A medium and high correlation was defined by $\alpha = 0.3$ and $\alpha = 0.01$, respectively (Eq. 24). The dashed black line describes the results obtained under the single-patch model. For each scenario, the mean was calculated across 1000 realizations of the stochastic model

overall population persistence, depending on the mean value of carrying capacity and the dispersal rate. For landscapes with an established mosquito population, a low mean patch carrying capacity and a low dispersal rate, increased spatial heterogeneity resulted in increases in total population size and patch occupancy compared with a spatially homogeneous landscape (Fig. 5a, b). This is because a small fraction of patches now have carrying capacities high enough to sustain local population persistence for an extended period. However, for higher values of mean patch carrying capacity, inter-patch variability increases the risk of stochastic extinction compared with a homogeneous landscape (Fig. 5a, b) — due to a higher proportion of patches having carrying capacities too low to allow local populations to persist.

Population dynamics also showed non-monotonicity in relation to the dispersal rate (Fig. 6a, b). For population invasions, the speed of invasion increased monotonically with the dispersal rate, but decreased with increasing between-patch variability (Fig. 6a). For established populations, with low between-patch variability, local persistence and overall population size was highest for high dispersal rates (Fig. 6b). However, increasing between-patch variability resulted in reduced overall population size, with population size lowest for high dispersal rates due to dispersal-driven depopulation of patches with high carrying capacity (Fig. 6b).

More realistic heterogeneous landscapes tend to show a high degree of local spatial correlation in carrying capacity, namely patches with similar levels of carrying capacity are clustered

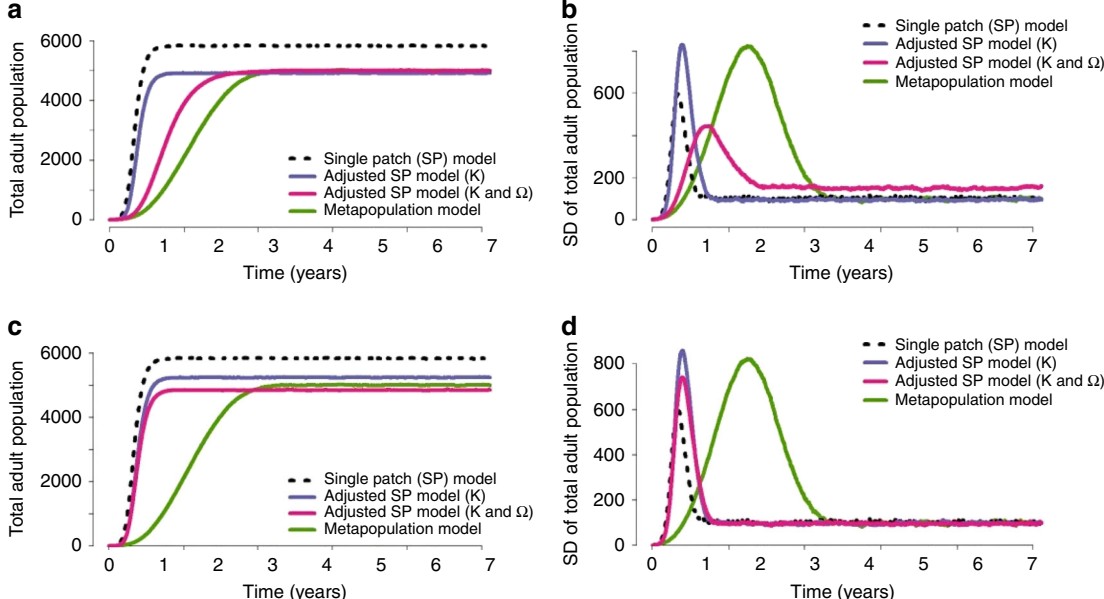

**Fig. 7** Single-patch approximations example. Comparison of results obtained when we approximate the invasion dynamics observed when a homogeneous landscape comprised 1024 patches, with an equilibrium larval population of 6 larvae per patch, is seeded with 6 adult mosquitoes. A mean dispersal length of one patch and a dispersal rate of 0.10 was used. **a**, **b** This corresponds to the scenario where we approximate two quantities — the mean equilibrium total adult mosquito population and the growth rate of the population. **c**, **d** Here, we approximate three quantities — the mean and variance of the equilibrium total adult mosquito population and the growth rate of the population. For each scenario, the mean and variance calculated across 1000 realizations of the stochastic model

together. Simulating dynamics on such landscapes, increasing spatial correlation resulted in increases in the mean equilibrium population size (Fig. 6c) but decreases in mean patch occupancy (Fig. 6d). This arises from improved local population persistence in areas with high patch-carrying capacity, but poorer local persistence in areas of low carrying capacity. Longer range dispersal tends to result in lower mean population sizes (Fig. 6c) but higher levels of patch occupancy (Fig. 6d) due to more frequent dispersal from high carrying capacity to low carrying capacity areas.

**Single-patch approximations.** We explored whether the invasion dynamics observed under the metapopulation model could be approximated by varying mean larval-carrying capacity ($\bar{K}$) and the power on density dependence ($\Omega$) in the corresponding single-patch model. To approximate the mean equilibrium total adult mosquito population size and growth rate observed under the metapopulation model, we first allowed $\bar{K}$ alone to vary while keeping $\Omega = 1$ fixed (i.e., assuming density dependence remained linear). Reducing mean larval-carrying capacity enabled us to closely approximate equilibrium population size (Fig. 7a). However, owing to the rapid speed of population growth in the single-patch model, this approach led us to overestimate the growth rate of the population and underestimate the variability in population size during the early stages of population growth (Fig. 7 and Supplementary Figs. 1 and 2).

Instead, allowing both $\bar{K}$ and $\Omega$ to vary resulted in a better approximation of the growth rate of the population (Fig. 7a, b and Supplementary Figs. 1a–d and 2a–d), with reduced $\bar{K}$ and $\Omega$ giving the best-fitting combination of values for fragmented landscapes where individual patches have a low carrying capacity (Supplementary Fig. 3). This is because $\Omega < 1$ (giving sub-linear density dependence) and reduced $K$ results in higher larval mortality and thus slower population growth, when larval density is low during the early stages of population growth following seeding, compared with single-patch models where $\Omega = 1$ and $K$

is larger. This in turn allowed us to better replicate the full temporal curve of population growth observed under the metapopulation model.

However, we remained unable to capture the increase in variability during the early stages of population growth (Fig. 7b and Supplementary Figs. 1c, d and 2c, d). Furthermore, in some cases where the carrying capacity of individual patches was low, this led to an overestimate of the variance of the equilibrium adult population (Fig. 7b and Supplementary Fig. 2c). Approximating the variance of the equilibrium adult population, in addition to the mean equilibrium adult population size and the growth rate, improved our estimate of the variance of the population but at the cost of then overestimating the speed of growth (Fig. 7c, d and Supplementary Figs. 1e–h and 2e–h). Similar patterns were observed when approximating the mean and variance of the equilibrium adult population only and not taking account of the growth rate.

As expected, the largest adjustments to the single-patch model were needed for fragmented landscapes where individual patches had a very low carrying capacity and where dispersal beyond the seeded patch was limited (Supplementary Fig. 3). In general, smaller adjustments to the single-patch model were required to approximate the dynamics in fragmented landscapes where changes to parameter values such as increasing the dispersal rate or mean dispersal length resulted in faster population growth and increased population size in the metapopulation model.

## Discussion
Mathematical models of dengue and malaria, which consider larval population dynamics, largely treat larval populations as a well-mixed population coming from a single breeding site, and thus do not account for the fragmented structure of larval populations. Here, by adopting a metapopulation approach to model the dynamics of mosquito populations, we examined the impact of larval habitat fragmentation and the role of the spatial structure in shaping mosquito population dynamics at fine spatial

scales. We found that fragmentation of larval populations may lead to a reduction in population size and patch occupancy (compared with a single well-mixed population), with the largest reductions occurring when individual patches had low carrying capacities and thus a substantial risk of stochastic population extinction. Both the dispersal behaviour of adult mosquitoes and features of the underlying landscape played key roles in driving the dynamics observed, through both counteracting and amplifying the effects of fragmentation. Moreover, we found that the population dynamics observed were heavily dependent on the level of spatial granularity represented in models, as reductions in granularity reduced the risk of extinction through stochastic effects, and hence increased population persistence, growth and patch occupancy across the landscape.

These results suggest that using non-spatial models to represent the dynamics of mosquito populations may substantially underestimate the stochastic volatility of those populations and the frequency at which local mosquito populations go extinct. However, these effects are greatest for low patch-carrying-capacity and relatively local adult mosquito dispersal. Assessing whether such effects are relevant to real-world mosquito populations therefore requires consideration of what is the appropriate level of representation of spatial structure in mosquito populations. The results obtained here suggest that, to capture the impact of larval habitat fragmentation on fine-scale mosquito population dynamics in a meaningful way, models should aim to capture the population dynamics observed when modelling at a level of spatial granularity such that movement between patches is representative of the typical dispersal length of the mosquito. If granularity is reduced such that movement between patches goes beyond the typical dispersal length, the vulnerability of small local populations to extinction is masked by the increase in patch size and consequent reduction in the risk of extinction through stochastic effects, and thus we fail to observe the potential impact of habitat fragmentation when populations are small.

As typical dispersal length varies between species of mosquito, the choice of scale will therefore vary according to disease being modelled. Mark–release–recapture (MRR) experiments suggest the dispersal range of *A. aegypti* mosquitoes is in the range 20–100 m[48,49], with most released *A. aegypti* being recaptured in the house of release[49]. Thus, modelling at the level of household would seem appropriate for that species. *A. aegypti* populations in dengue-endemic areas range between under 0.5 to over 3 adult females per person — namely in the range 2 to 20 per household, depending on household size[26,50], with empirical evidence to suggest considerable spatial heterogeneity in vector population density at the individual household level[25,26,33,34,50,51]. Thus, the effects of local dispersal and landscape fragmentation described here may be highly relevant to the modelling of *A. aegypti* — and thus dengue, Zika and Chikungunya — as mosquito densities at the household level can drop to very low numbers during seasonal troughs[31].

However, measured mean dispersion distances are an order of magnitude further for *Anopheles gambiae* — 500 m or greater[52,53] — meaning appropriate patch sizes (both in terms of dimension and population size) can be expected to be considerably larger, perhaps village level. Thus, the effects of landscape heterogeneity, local stochastic extinction and re-seeding predicted by our modelling to be significant for *A. aegypti* are likely to be less so for *A. gambiae*. The exception might be in environments where *A. gambiae* density is exceptionally low — e.g., during seasonal troughs in areas such as the Sahel[54].

As density-dependent competition gives rise to nonlinear dynamics during the larval stage of population growth and the risk of extinction through stochastic effects is highest for small local populations, the dynamics observed when the fragmented structure of larval populations is explicitly represented cannot be exactly reproduced using a single-patch (non-spatial) model. In a regime where local stochastic effects are substantial, total mosquito population size will be smaller (and population volatility larger) than predicted by a single-patch model with the same total carrying capacity.

However, carrying capacity is an unmeasurable and quite theoretical concept. More typically, modellers calibrate models to reproduce observed population characteristics, such as population size. In that context, our results indicate that single-patch models will underestimate population volatility and overestimate invasion speeds (e.g., following population troughs). Interestingly, we found that stochastic volatility can be better captured by a single-patch model if the intensity of density dependence is assumed to be less than linear, but even this adjustment is unable to capture the slower invasion dynamics and increased variability associated with the spatial structuring of the mosquito population.

These fine-scale dynamics are likely to be of particular importance when modelling the potential impact of novel biological vector control measures, such as *Wolbachia*-infected *A. aegypti*[55] or homing endonuclease genes[56]. As these methods rely on the successful establishment and propagation of modified mosquito populations across a landscape, a strong understanding of mosquito population dynamics at fine spatial scales is critical to understanding the likelihood of successful establishment, persistence and spread. Underestimating the stochastic volatility of mosquito populations and failing to consider the role of the underlying spatial structure may make it more difficult to identify potential barriers to successful establishment and spread, and thus to estimate the likely impact of these strategies on disease dynamics.

The work presented here however has a number of limitations. As we consider the dynamics of mosquito populations alone, the extent to which our results may affect models of dengue and malaria transmission, which consider larval population dynamics, is yet to be explored. A model that incorporates both vector and disease dynamics would offer much greater insight into the potential impact of larval population fragmentation on the fine-scale transmission dynamics of dengue and malaria. We have also assumed that larval development times and the lifespan of adult mosquitoes are exponentially distributed. However, the rate at which mosquito larvae develop is temperature dependent[19,57] and older adult mosquitoes may experience increased mortality[58,59]. A further limitation is that we have only studied random landscapes here. Work is needed to parameterize spatially explicit models of mosquito dynamics against data from real landscapes, such as cities (e.g., Yamashita et al.[60] who deterministically modelled the influence of urban landscapes and wind on mosquito dispersion). Last, although appropriate values for model parameters were sourced where possible from the existing literature, the model presented was not explicitly fitted to entomological data. Future work will build on the work presented here by addressing these limitations, with the aim of providing a more complete picture of the impact of larval habitat fragmentation on the persistence and control of dengue and malaria at fine spatial scales.

In conclusion, our study highlights the importance of choices made concerning representations of the spatial structure of larval populations when modelling the dynamics of mosquito populations at fine spatial scales. Our results demonstrate that, for low mosquito population densities, adopting a single-patch approach to model the fine-scale dynamics of larval populations may substantially underestimate the stochastic volatility of mosquito populations and overestimate the speed of population invasion and growth. Accounting for the fragmented structure of larval populations allows us to better capture the dynamics of

density-dependent competition and therefore to understand how the level of connectivity between local populations and features of the underlying landscape drive the population dynamics observed at fine spatial scales.

## Methods

**Model structure**. We developed a fine-scale stochastic metapopulation model of mosquito population dynamics, where patches are arranged in an $n \times n$ grid and each individual patch $(i, j)$ represents a local mosquito population ($1 \leq i, j \leq n$). Local populations comprised an egg, larval and a female adult mosquito population, and are connected through adult mosquito dispersal, where adult mosquitoes can move to and lay eggs in neighbouring patches. The deterministic dynamics of a local population in patch $(i, j)$ are described by the following set of equations:

$$\frac{dE_{ij}(t)}{dt} = bgO_{ij}(t) - \gamma_E E_{ij}(t) - \mu_E E_{ij}(t) \tag{1}$$

$$\frac{dL_{ij}(t)}{dt} = \gamma_E E_{ij}(t) - \gamma_L L_{ij}(t) - \mu_L \left(1 + \left(\frac{L_{ij}(t)}{K_{ij}(t)}\right)^\Omega\right) L_{ij}(t) \tag{2}$$

$$\frac{dA_{ij}(t)}{dt} = M_{ij}(t) + \gamma_L L_{ij}(t) - \mu_A A_{ij}(t) \tag{3}$$

$$O_{ij}(t) = g^{-1} A_{ij}(t) \tag{4}$$

$$M_{ij}(t) = r\left(\sum_{(i'j')} A_{i'j'}(t) q(d_{(ij)-(i'j')}) - \sum_{(i'j')} A_{ij}(t) q(d_{(ij)-(i'j')})\right) \tag{5}$$

Here, $E_{ij}(t)$, $L_{ij}(t)$ and $A_{ij}(t)$ denotes the egg, larval and adult population in $(i, j)$ at time $t$, respectively; $K_{ij}(t)$ denotes the larval-carrying capacity of $(i, j)$ at time $t$; $O_{ij}(t)$ denotes the number of adult mosquitoes laying eggs in $(i, j)$ at time $t$; $M_{ij}(t)$ denotes the net migration of adult mosquitoes into $(i, j)$ at time $t$; $\Omega$ describes the power on density dependence (with $\Omega = 1$ describing linear density dependence); $b$ denotes the oviposition rate; $\gamma_E$ and $\gamma_L$ denote the egg and larval development rate, respectively; $g$ denotes the length of the gonotrophic cycle of adult female mosquitoes. $\mu_E$, $\mu_L$ and $\mu_A$ denote the egg, larval and adult mosquito mortality rate, respectively.

For each patch $(i, j)$, we represent density-dependent competition during the larval stage of development through increasing larval mortality as the size of the larval population approaches the carrying capacity of the patch. Although density-dependent competition could have been represented by varying the oviposition or larval development rate in accordance with larval density (instead of or in addition to the larval mortality rate), we chose to represent this solely through changes in larval mortality in accordance with changes in larval population density, as this approach is most commonly used in models of mosquito population dynamics, which consider larval populations[39,41–43,61].

The basic mosquito reproduction number, $R_M$, is defined as the average number of adult females produced by a single female mosquito during her lifespan, in the absence of population regulation. For our model, $R_M$ is given by

$$R_M = b \frac{\gamma_E}{\gamma_E + \mu_E} \frac{\gamma_L}{\gamma_L + \mu_L} \frac{1}{\mu_A} \tag{6}$$

Here, $\frac{b}{\mu_A}$ describes the average number of eggs laid by a female over the course of her lifetime, whereas the terms $\frac{\gamma_E}{\gamma_E + \mu_E}$ and $\frac{\gamma_L}{\gamma_L + \mu_L}$ describe the probability of survival during the egg and larval stages, respectively, when determining the average number of females produced. Here, the value of $b$ in Eqs. (1)–(5) is assigned so that $R_M$ remains fixed.

Equation 5 above describes the dispersion of adult mosquitoes, where $r$ is the dispersal rate and $q(d)$ is the probability of moving distance $d$ if dispersal occurs, otherwise known as the dispersion kernel. Here, $d_{(ij)-(i'j')}$ is the distance between the centroids of patches $(i, j)$ and $(i', j')$. We set $d_{(ij)-(ij)} = 0.5214$ (the average distance travelled between two random points in a unit square). We use a discretized version of a continuous space negative exponential kernel with a maximum dispersion distance $D$, defined as:

$$q(d) = \begin{cases} Q e^{\left(-\frac{d}{a}\right)}, & \text{for } d \leq D \\ 0 & \text{for } d > D \end{cases} \tag{7}$$

where $Q$ is a normalization constant defined as

$$Q = \frac{1}{\max_{(i,j)}\left\{\sum_{(i'j')|d_{(ij)-(i'j')}<D} e^{\left(-\frac{d_{(ij)-(i'j')}}{a}\right)}\right\}} \tag{8}$$

and $m = 2a$ describes the mean dispersal length. Mosquitoes are assumed to not disperse outside the modelled population; therefore, total net dispersion rates are lower from patches at the edge of the modelled space than from interior patches.

We implemented a discrete time stochastic version of this model. In a time step of one eighth of a day $\delta t = 0.125$, we draw $O_{ij}(t)$, the number of adult mosquitoes

laying eggs in patch $(i, j)$ at time $t$, from a Binomial distribution

$$O_{ij}(t) \sim Bin(A_{ij}(t), g^{-1}) \tag{9}$$

and $N_{ij}^E(t)$ the number of eggs laid in $(i, j)$ at time $t$, from a Poisson distribution

$$N_{ij}^E(t) \sim Poisson(bgO_{ij}(t)\delta t) \tag{10}$$

We then determine $N_{ij}^L(t)$ and $D_{ij}^E(t)$, the number of new larvae and deaths during the egg stage in $(i, j)$ at time $t$, respectively, using a competing hazards model as follows:

$$h_{ij}^E(t) = \gamma_E + \mu_E \tag{11}$$

$$T_{ij}^E(t) \sim Bin(E_{ij}(t), h_{ij}^E(t) \cdot \delta t) \tag{12}$$

$$N_{ij}^L(t) \sim Bin\left(T_{ij}^E(t), \frac{\gamma_E}{h_{ij}^E(t)}\right) \tag{13}$$

$$D_{ij}^E(t) = T_{ij}^E(t) - N_{ij}^L(t) \tag{14}$$

where $h_{ij}^E(t)$ describes the total hazard of leaving the egg population in $(i, j)$ at time $t$, and $T_{ij}^E(t)$ denotes the total number of eggs leaving $(i, j)$ at time $t$.

We use a similar competing hazards approach to determine (i) the number of new adult mosquitoes and larval deaths in $(i, j)$ at time $t$ and (ii) the number of adult mosquito deaths and the number of adult mosquitoes dispersing to patches at each distance within the dispersal range. The total hazard of leaving the larval population in $(i, j)$ at time $t$ is given by

$$h_{ij}^L(t) = \gamma_L + \mu_L \left(1 + \left(\frac{L_{ij}(t)}{K_{ij}(t)}\right)^\Omega\right) \tag{15}$$

The total hazard of leaving the adult population in $(i, j)$ at time $t$ is given by

$$h_{ij}^A(t) = \sum_d g_{ij}(d) + \mu_A = r + \mu_A \tag{16}$$

where $g_{ij}(d)$ denotes the the hazard (with respect to an individual adult mosquito) of dispersing to a patch at distance $d$ from $(i, j)$. This is given by

$$g_{ij}(d) = rq(d)n_{ij}(d) \tag{17}$$

where $r$ denotes the dispersal rate, $n_{ij}(d)$ denotes the number of patches at distance $d$ from patch $(i,j)$ and $q(d)$ denotes the probability of moving distance $d$, if dispersal occurs. For each adult mosquito dispersing a distance $d$ from $(i, j)$, its destination patch is randomly selected from among the $n_{ij}(d)$ neighbours at this distance.

**Spatial granularity and parameter values**. We vary the level of fidelity we have in our model in representing the true underlying fragmented structure of larval mosquito populations by varying the number of patches used to represent the dynamics of the mosquito population across a given landscape, with more patches corresponding to greater fidelity. The lowest level of spatial granularity in our model corresponds to the single patch approach, as here we do not account for the fragmented structure of larval populations and consider the population as a single, well-mixed population coming from one large breeding site. Finer levels of spatial granularity allow us to account for the fragmented structure of larval populations and spatially heterogeneous mixing among populations, at different levels.

Parameter values were chosen where possible from the literature to represent the characteristics of A. aegypti. Hence, in choosing the finest level of spatial granularity in our model, we considered the typical oviposition and dispersal behaviour of adult female A. aegypti.

MRR studies indicate that, owing to their domestic habitat, A. aegypti typically disperse very small distances (e.g., 20–100 m[49], 56 m[62]), often only travelling to neighbouring households, with most mosquitoes released in individual households during MRR studies recaptured within the house of release[49]. Thus, mixing of A. aegypti among households is spatially heterogeneous[25,26,63].

Within individual households, mixing among larval populations is also dependent on the oviposition behaviour of adult females. A. aegypti often lay eggs from a single batch across multiple breeding sites (skip-oviposition)[30,63], and thus may distribute their eggs across a household. However, empirical field studies indicate that adult female A. aegypti actively choose oviposition sites and thus do not distribute eggs randomly among breeding sites[30,63]. Analysis of the egg-laying behaviour of A. aegypti suggests that breeding site selection is driven by a variety of factors including conspecific attraction[30], food resources[30], and the physical properties of individual breeding containers[63]. Therefore, given the variety of factors that contribute towards oviposition site selection within an individual household, we make the simplifying assumption that mixing within individual households is spatially homogeneous as the motivation for this study is not to develop a highly detailed model of A. aegypti population dynamics at the individual breeding container level. Rather, we aim to explore how allowing different levels of spatial heterogeneity in mixing among larval populations, and our choice of representation of spatial structure (if represented at all) influence our understanding of mosquito population dynamics at fine spatial scales. Thus, we

**Table 1 Model parameter values**

| Parameter | Definition | Value | Refs |
|---|---|---|---|
| $g$ | Length of gonotrophic cycle | 3 Days | 66 |
| $R_M$ | Mosquito reproduction number | 2.69 | 67 |
| $b$ | Daily oviposition rate | Assigned to match $R_M$ | – |
| $\frac{1}{\gamma_E}$ | Mean development time of mosquito eggs | 4 Days | 19 |
| $\gamma_L$ | Mean development time of mosquito larvae | 15 Days | 19 |
| $\mu_E$ | Egg mortality rate | 0.01 Day$^{-1}$ | 61 |
| $\mu_L$ | Larval mortality rate | 0.025 Day$^{-1}$ | 61 |
| $\mu_A$ | Adult mosquito mortality rate | 0.1 Day$^{-1}$ | 68 |
| $m$ | Mean dispersal length of an adult mosquito | 5 Patches day$^{-1}$ | 48,49,62 |
| $D$ | Maximum dispersal range of an adult mosquito | 12 Patches day$^{-1}$ | 48,49,62 |
| $r$ | Adult mosquito dispersal rate | 0.08 Day$^{-1}$ | – |
| $\bar{K}_{ij}$ | Mean-larval carrying capacity of patch $(i, j)$ | Variable | – |
| $\Omega$ | Power on density dependence | 1 | – |

Definition and values of parameters used in the simulation model. A negative exponential dispersal kernel with a mean dispersal length of 5 was used and a dispersal range of 12 was chosen to aid computational efficiency

chose the finest level of spatial granularity in our model to correspond to modelling at the individual household level, as movement between patches at this level of granularity is characteristic of the typical dispersal length of *A. aegypti*.

We consider an example urban landscape comprising 1024 households. Thus, at the finest level of spatial granularity in our model, we have a $32 \times 32$ spatial grid. We assume each square of this $32 \times 32$ grid represents an area of ~20 m × 20 m. We coarsen the level of spatial granularity in our model by doubling the size of each patch. Hence, as we move from a $32 \times 32$ grid to a $16 \times 16$ grid, each patch now corresponds to a group of four households and an area of 40 m × 40 m. We examine the effect of further coarsening the representation of space in our model until we represent the dynamics of the *A. aegypti* population across the landscape using a single patch, where the population is assumed to mix homogeneously.

**Parameter values.** Unless otherwise stated, all parameter values used are as shown in Table 1. Our choice of parameter values used to represent the typical dispersal behaviour of *A. aegypti* at the individual household level was guided by estimates of the mean dispersal distance derived from MRR studies. As discussed above, MRR studies indicate that *A. aegypti* typically disperse very short distances[48,49,62]. Although estimates of the mean dispersal distance vary between studies and locations, a distance of ~50 m is typical of values the mean dispersal distance of *A. aegypti* estimated by MRR studies[48,49,62].

These estimates however largely correspond to the mean distance travelled over the course of an adult female's lifespan, rather than over the course of a single day. Hence, we sought to choose values of the daily dispersal rate and mean dispersal length (at the $32 \times 32$ grid level) to correspond to a mean lifetime dispersal distance of ~50 m. We chose these parameter values by first using our model to explore the range of mean lifetime dispersal distances generated by combinations of the daily dispersal rate ($r$) and mean dispersal length ($m$) (keeping the adult mosquito mortality rate ($\mu_A$) fixed).

To calculate the mean lifetime dispersal distance for a given combination of the daily dispersal rate and mean dispersal length in model run $n$, we first set the oviposition rate ($b$) to zero and seed 1000 adult females in the same patch ($i', j'$) at the centre of our $32 \times 32$ grid. Setting the oviposition rate to zero ensures that no new adult females enter the population. We then record the cumulative number of deaths in each patch on the grid at the end of the run and calculate the Euclidean distance between each patch and the seeded patch. The mean lifetime dispersal distance for run $n$ ($MLD_n$) is then given by

$$\text{MLD}_n = \frac{1}{1000} \sum_{(i,j)} C_{ij}(n) d_{(ij)-(i'j')} \qquad (18)$$

where $C_{ij}(n)$ denotes the cumulative number of adult female deaths in patch $(i, j)$ at the end of run $n$ and $d_{(ij)-(i'j')}$ denotes the distance between patch $(i, j)$ and the seeded patch $(i', j')$. To get a final estimate of the mean lifetime dispersal distance, we take the average across 1000 model runs (seeding in the same patch in each run).

The results of this exercise are presented in Supplementary Fig. 4. We found that, using a negative exponential kernel, a daily dispersal rate of 0.08 and mean dispersal length of 5 patches per day gave a mean lifetime dispersal distance of ~50 m. Hence, we chose these values as the default parameter values in our model. However, we tested sensitivity of our modelling results to different values of these parameters to explore how the underlying population dynamics are affected by different dispersal behaviours.

As the level of granularity in our model is reduced, we adjust the daily mean and maximum dispersal lengths accordingly. For example, as we move from a $32 \times 32$ grid to a $16 \times 16$ grid, we double the size of each patch. Thus, the mean and maximum dispersal lengths are halved.

**Landscape model.** Larval-carrying capacity is varied seasonally using a sinusoidal function of the form

$$K_{ij}(t) = \bar{K}_{ij} \left[ 1 + \Delta \left( \cos \left( 2\pi \left( \frac{t\delta t}{365} + \phi \right) \right) \right) \right] \qquad (19)$$

where here $K_{ij}(t)$ denotes larval carrying capacity at time $t$, $\Delta$ and $\phi$ denote the amplitude and phase of seasonal variation in carrying capacity, respectively, and $\bar{K}_{ij}$ denotes the mean carrying capacity of patch $(i, j)$ across the year.

Spatial homogeneous landscapes (Fig. 8a) are created by assuming that each patch had the same initial population, and that the local population is at its deterministic equilibrium at the start of each simulation. $\bar{K}_{ij}$ is thus set as

$$\bar{K}_{ij} = \frac{L_{ij}^*}{\frac{\gamma_L}{\mu_L} \left( \frac{b\gamma_E}{\mu_A(\gamma_E + \mu_E)} - 1 \right) - 1} \qquad \forall i, j \qquad (20)$$

where $L_{ij}^*$ denotes the larval population of $(i, j)$ at deterministic equilibrium.

To create a spatially heterogeneous landscape (Fig. 8b) with a fixed mean larval-carrying capacity $\bar{K}^*$ and a specified level of variability $\sigma^2$ around this mean value, we first draw a sequence of positive values $X_{ij}$ from a log-normal distribution

$$X_{ij} \sim \log N(\bar{K}^*, \sigma^2) \qquad (21)$$

To keep the total carrying capacity across the landscape fixed, we apply a transformation of the form $aX_{ij}^b$ to the generated values, where values of $a$ and $b$ are such that

$$\mathbb{E}[aX_{ij}^b] = \bar{K}^* \qquad (22)$$

$$\text{Var}[aX_{ij}^b] = \sigma^2 \qquad (23)$$

and set $\bar{K}_{ij} = aX_{ij}^b$. Values of $a$ and $b$ such that (22) and (23) hold are found numerically using linear interpolation.

To vary the level of spatial correlation in values of carrying capacity across a heterogeneous landscape (Fig. 8c), we adopt a similar approach to that used elsewhere by Hancock et al.[64]. An $n^2 \times n^2$ correlation matrix $C$ with entries

$$C_{ij} = e^{-\alpha d} \qquad (24)$$

is created, where $d$ denotes the distance between a pair of patches and $\alpha$ controls the degree of correlation between values, with very small values of $\alpha$ giving a high level of correlation. Taking the Cholesky decomposition of $C$ gives a matrix $L$ such that $C = LL^T$, and hence for $X \sim N(0, I)$, we have that $\text{Corr}[L\tilde{X}] = C$[65]. Setting $Y = e^{LX}$, a similar approach to that described directly above by Eqs. (22) and (23) is then used to transform the values of $Y$ to give a landscape with a mean larval-carrying capacity $\bar{K}^*$ and variance $\sigma^2$.

We consider both the population dynamics for landscapes where a stable mosquito population is already established and the invasion dynamics resulting from mosquitoes being seeded into an otherwise unoccupied landscape. For all model runs and for each landscape type, the seeded patch was chosen at random from among the patches on our $32 \times 32$ spatial grid. We evaluate the effects of spatial structure and heterogeneity on population size, population persistence, the level of patch occupancy and speed of population spread across the landscape.

**Single patch approximations.** We explored whether the invasion dynamics observed for a homogeneous landscape under the metapopulation model at the individual household level ($32 \times 32$ grid) could be approximated by varying parameters in the corresponding single-patch model whose values are not constrained by directly observed ecological processes — namely mean larval-carrying

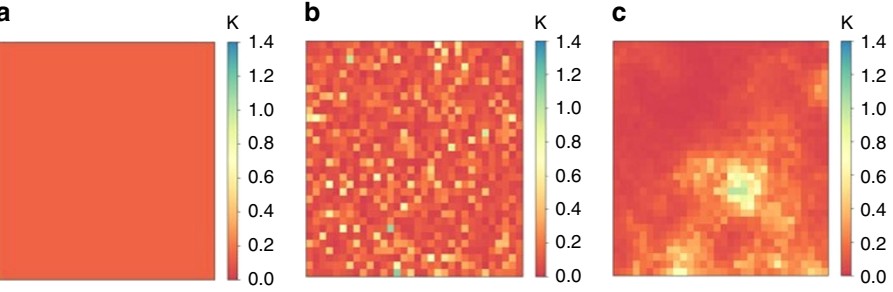

**Fig. 8** Example landscapes. Example of a landscape with a fixed total and mean carrying capacity: **a** homogeneous landscape, **b** heterogeneity in carrying capacity without clustering, and **c** heterogeneity in carrying capacity and a high level of clustering across the landscape ($\alpha = 0.01$)

capacity ($\bar{K}$) and the power on density-dependent larval mortality ($\Omega$). Hence, by varying $\Omega$, we allow for nonlinear changes in the larval mortality rate as the larval population grows. We then compared the results obtained using the metapopulation model and adjusted single-patch model and adopted a least-squares approach to determine the combination of values of $\bar{K}$ and $\Omega$ best approximating the metapopulation dynamics. We sought to approximate the mean and variance of the equilibrium total adult mosquito population and the time until the population reaches equilibrium (a proxy for the growth rate of the population).

More formally, for a homogeneous landscape where $L_{ij}^*$ denotes the deterministic equilibrium larval population of each patch $(i, j)$ in the metapopulation model, we allow $\bar{K}$ and $\Omega$ to vary in the corresponding single-patch model (irrespective of the value of $L_{ij}^*$ and keeping all other parameter values fixed), denoting these varying parameters $\bar{K}'$ and $\Omega'$. To approximate the mean and variance of the equilibrium total adult mosquito population observed under the metapopulation model for example, we define the sum of squared errors associated with the approximation, $\epsilon(L_{ij}^*, \bar{K}', \Omega')$, as

$$\epsilon(L_{ij}^*, \bar{K}', \Omega') = \left[\frac{M(\bar{K}', \Omega') - M(L_{ij}^*)}{M(L_{ij}^*)}\right]^2 + \left[\frac{V(\bar{K}', \Omega') - V(L_{ij}^*)}{V(L_{ij}^*)}\right]^2 \quad (25)$$

where $M(L_{ij}^*)$ and $M(\bar{K}', \Omega')$ denote the mean equilibrium total adult mosquito population size observed under the metapopulation model and adjusted single-patch model, respectively, and $V(L_{ij}^*)$ and $V(\bar{K}', \Omega')$ denote the variance of the equilibrium total adult mosquito population observed under the metapopulation model and adjusted single-patch model, respectively. The combination of $(\bar{K}', \Omega')$, which minimize this error, is selected as the best-fitting values of $\bar{K}'$ and $\Omega'$.

**Reporting summary**. Further information on research design is available in the Nature Research Reporting Summary linked to this article.

## Data Availability
No datasets were generated or analysed during the current study.

## Code Availability
The source code for the metapopulation model can be found at: https://github.com/claremccormack/Metapop_Model

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

## Acknowledgements

We acknowledge grant funding from the Bill and Melinda Gates Foundation, the NIGMS MIDAS programme and joint Centre funding from the UK Medical Research Council and Department for International Development (grant MR/R015600/1). C.M.C. also thanks the UK Medical Research Council for PhD studentship funding (grant MR/L501414/1).

## Author contributions

Conceived the study: C.M.C., A.C.G. and N.M.F. Performed the analysis: C.M.C. Wrote the initial draft of the manuscript: C.M.C. All authors reviewed and approved the final manuscript.

## Additional information

**Competing interests:** The authors declare no competing interests.

