## [Peer Review File · Communications Biology]

Reviewers' comments:

Reviewer #1 (Remarks to the Author):

Most mathematical models of mosquito abundance treat the mosquito population as one well-mixed patch, in spite of field evidence that suggests meta-populations are highly clustered. The authors use a stochastic meta-population model to explore how changes in patch size, dispersal rate, larval carrying capacity, and spatial autocorrelation impact mosquito dynamics. They find that a single-patch models overestimate equilibrium population sizes, although the magnitude of this is dependent on the four factors listed above. Interestingly, while adjusting larval carrying capacity and density dependence, allowed a single patch model to accurately estimate the equilibrium population size, it continued to overestimate the growth rate.

Strengths:

This study is well-motivated and very relevant given the ever increasing number of vector-borne disease models being published.

This is a very thorough modeling study that, in spite of the number of factors studied, very clearly lays out its approach and findings. The results, while not completely unexpected, explore non-linearities and interactions between landscape and mosquito factors that would not be immediately apparent otherwise.

I especially appreciate the concrete recommendations for incorporating patch dynamics into models on lines 248-250.

Minor weaknesses:

Methods:

What was the unit of time for your model? Your parameters are in days, but it is never explicitly said, and the mention of the seasonality based on years (also using t) may confuse the reader.

The way you created differently autocorrelated landscapes is very clever and clearly described (lines 481-502).

How many times were your models simulated? Some of the supplemental figures mention 1000, but it is never mentioned in the text. If it differs for each approach, it could be added to figure legends as well.

Line 504: It would be helpful to have more information on how invasions were seeded. Were all mosquitoes seeded into the same patch? In the case of heterogeneous landscapes, were simulations rerun with different initial seeding locations to get an average for that landscape? I think the placement of seeding could be especially important given the edge effects you mention.

Line 508: Was this a comparison between a single patch model and a homogenous multi-patch model? What was the granularity?

Results:

Line 201: It seems counter-intuitive that reducing the strength of density-dependent larval mortality would decrease the population growth rate. It would be helpful to the reader to clarify again the role

of Omega in your equations and that this isn't a linear decrease in density-dependence. This could also be mentioned in the methods (line 338).

Figure 3d: legend text is overlapping, making number unreadable.

Figure 4: Please double-check that this figure legend is correct. It seems like a shorter dispersal distance would result in c/d, not a longer one, based on the text directly above.

Figure 7: The legend for this figure is slightly confusing. What is the difference between a/b and c/d? The legend suggests that c is the variance of the total population at each time step. Is the variance meant to be across simulations? I also am not sure that c is a depiction of variance. Please double-check these labels are correct.

In general, it would be wise to go through and ensure all of your figure legends are correct, as some seem to have panels mixed up in the legends.

Line 198: Mislabeled figure reference.

Reviewer #2 (Remarks to the Author):

Dear Editor,

This paper presents an innovative mathematical model describing mosquito population dynamics considering spatial distribution of patches with different carrying capacity. Simulations using this model were carried out evidencing that spatial distribution of patches can modify significantly the final results on the mosquito population dynamics.

1. General critics over the presented model is that it relays on tens of parameters and particular distributions. It is not trivial how to obtain these data from experiments and thus how to use this model as a quantitative study in the public health management. However, the model is perfectly valid for qualitative investigations.
2. On the page 17 the basic mosquito reproduction number R is defined as a product of three terms. The two containing "gamma" represent the survivor probability and not mortality as stated in the text. The equation itself is correct. Also, it is not clear how the term R appears in the system (1)-(5).
3. Please pay attention to the punctuation in the equations (15), (16) and (17) on page 18.
4. Mathematical models dealing with spatial dependence of carrying capacity and the impact on the mosquito population dynamics were addressed in other works. The authors may be interested in comparing the presented methodology with one proposed in Parasites & Vectors 2018, 11:245 (<https://doi.org/10.1186/s13071-018-2829-1>).

In view of mentioned remarks, the presented study is new and interesting for the community.

Reviewers' comments:

Reviewer #1 (Remarks to the Author):

Most mathematical models of mosquito abundance treat the mosquito population as one well-mixed patch, in spite of field evidence that suggests meta-populations are highly clustered. The authors use a stochastic meta-population model to explore how changes in patch size, dispersal rate, larval carrying capacity, and spatial autocorrelation impact mosquito dynamics. They find that a single-patch models overestimate equilibrium population sizes, although the magnitude of this is dependent on the four factors listed above. Interestingly, while adjusting larval carrying capacity and density dependence, allowed a single patch model to accurately estimate the equilibrium population size, it continued to overestimate the growth rate.

Strengths:

This study is well-motivated and very relevant given the ever increasing number of vector-borne disease models being published.

This is a very thorough modelling study that, in spite of the number of factors studied, very clearly lays out its approach and findings. The results, while not completely unexpected, explore non-linearities and interactions between landscape and mosquito factors that would not be immediately apparent otherwise.

I especially appreciate the concrete recommendations for incorporating patch dynamics into models on lines 248-250.

Minor weaknesses:

Methods:

What was the unit of time for your model? Your parameters are in days, but it is never explicitly said, and the mention of the seasonality based on years (also using t) may confuse the reader.

The way you created differently autocorrelated landscapes is very clever and clearly described (lines 481-502).

How many times were your models simulated? Some of the supplemental figures mention 1000, but it is never mentioned in the text. If it differs for each approach, it could be added to figure legends as well.

Line 504: It would be helpful to have more information on how invasions were seeded. Were all mosquitoes seeded into the same patch? In the case of heterogeneous landscapes, were simulations rerun with different initial seeding locations to get an average for that landscape? I think the placement of seeding could be especially important given the edge effects you mention.

Line 508: Was this a comparison between a single patch model and a homogenous multi-patch model? What was the granularity?

Results:

Line 201: It seems counter-intuitive that reducing the strength of density-dependent larval mortality would decrease the population growth rate. It would be helpful to the reader to clarify again the role of Omega in your equations and that this isn't a linear decrease in density-dependence. This could also be mentioned in the methods (line 338).

Figure 3d: legend text is overlapping, making number unreadable.

Figure 4: Please double-check that this figure legend is correct. It seems like a shorter dispersal distance would result in c/d , not a longer one, based on the text directly above.

Figure 7: The legend for this figure is slightly confusing. What is the difference between a/b and c/d ? The legend suggests that c is the variance of the total population at each time step. Is the variance meant to be across simulations? I also am not sure that c is a depiction of variance. Please double-check these labels are correct.

In general, it would be wise to go through and ensure all of your figure legends are correct, as some seem to have panels mixed up in the legends.

Line 198: Mislabeled figure reference.

Reviewer #2 (Remarks to the Author):

Dear Editor,

This paper presents an innovative mathematical model describing mosquito population dynamics considering spatial distribution of patches with different carrying capacity. Simulations using this model were carried out evidencing that spatial distribution of patches can modify significantly the final results on the mosquito population dynamics.

1. General critics over the presented model is that it relays on tens of parameters and particular distributions. It is not trivial how to obtain these data from experiments and thus how to use this model as a quantitative study in the public health management. However, the model is perfectly valid for qualitative investigations.
2. On the page 17 the basic mosquito reproduction number R is defined as a product of three terms. The two containing "gamma" represent the survivor probability and not mortality as stated in the text. The equation itself is correct. Also, it is not clear how the term R appears in the system (1)-(5).
3. Please pay attention to the punctuation in the equations (15), (16) and (17) on page 18.
4. Mathematical models dealing with spatial dependence of carrying capacity and the impact on the mosquito population dynamics were addressed in other works. The authors may be interested in comparing the presented methodology with one proposed in Parasites & Vectors 2018, 11:245 (<https://doi.org/10.1186/s13071-018-2829-1>).

In view of mentioned remarks, the presented study is new and interesting for the community.

Response to Reviewers' Comments

(Page numbers refer to revised manuscript)

Reviewer #1 (Remarks to the Author):

Most mathematical models of mosquito abundance treat the mosquito population as one well-mixed patch, in spite of field evidence that suggests meta-populations are highly clustered. The authors use a stochastic meta-population model to explore how changes in patch size, dispersal rate, larval carrying capacity, and spatial autocorrelation impact mosquito dynamics. They find that a single-patch models overestimate equilibrium population sizes, although the magnitude of this is dependent on the four factors listed above. Interestingly, while adjusting larval carrying capacity and density dependence, allowed a single patch model to accurately estimate the equilibrium population size, it continued to overestimate the growth rate.

Strengths:

This study is well-motivated and very relevant given the ever increasing number of vector-borne disease models being published.

This is a very thorough modelling study that, in spite of the number of factors studied, very clearly lays out its approach and findings. The results, while not completely unexpected, explore non-linearities and interactions between landscape and mosquito factors that would not be immediately apparent otherwise.

I especially appreciate the concrete recommendations for incorporating patch dynamics into models on lines 248-250.

We thank the reviewer for their kind comments.

Minor weaknesses:

Methods:

What was the unit of time for your model? Your parameters are in days, but it is never explicitly said, and the mention of the seasonality based on years (also using t) may confuse the reader.

We used a time step of one eighth of a day ($\delta t=0.125$) in model simulations. This has now been clarified in the main text (page 19), and equation 19 has been amended to reflect how seasonality is calculated at each time step rather than yearly (page 24).

"We implemented a discrete time stochastic version of this model. In a time step of one eighth of a day $\delta t=0.125$, we draw ..."

"Larval carrying capacity is varied seasonally using a sinusoidal function of the form

$$K_{ij}(t) = \overline{K_{ij}} \left[1 + \Delta \left(\cos \left(2\pi \left(\frac{t\delta t}{365} + \phi \right) \right) \right) \right]$$

where here $K_{ij}(t)$ denotes larval carrying capacity at time t , Δ and ϕ denote the amplitude and phase of seasonal variation in carrying capacity respectively, and $\overline{K_{ij}}$ denotes the mean carrying capacity of patch (i,j) across the year.”

How many times were your models simulated? Some of the supplemental figures mention 1000, but it is never mentioned in the text. If it differs for each approach, it could be added to figure legends as well.

For each scenario, 1000 simulations of the model were performed. This has now been clarified in all figure legends, both in the main text and supplementary information.

Line 504: It would be helpful to have more information on how invasions were seeded. Were all mosquitoes seeded into the same patch? In the case of heterogeneous landscapes, were simulations rerun with different initial seeding locations to get an average for that landscape? I think the placement of seeding could be especially important given the edge effects you mention.

We agree that the placement of seeding may be an important factor in determining the likelihood of population persistence. Thus, for each model run and for all landscape types, the seeded patch was chosen at random from the 32x32 patches on the spatial grid. Thus, our results for invasions of heterogeneous landscapes correspond to an average for that landscape.

Additional text has been included in the Methods section (page 26) clarifying how invasions were seeded.

*“We consider both the population dynamics for landscapes where a stable mosquito population is already established and the invasion dynamics resulting from mosquitoes being seeded into an otherwise unoccupied landscape. **For all model runs and for each landscape type, the seeded patch was chosen at random from among the patches on our 32x32 spatial grid.** We evaluate the effects of spatial structure and heterogeneity on population size, population persistence, the level of patch occupancy and speed of population spread across the landscape.”*

Line 508: Was this a comparison between a single patch model and a homogenous multi-patch model? What was the granularity?

The single patch approximations refer to a comparison between the single patch model and a homogeneous 32x32 multi-patch model (individual household level).

This has now been clarified in the main text (page 26).

*“We explored if the invasion dynamics observed **for a homogeneous landscape** under the metapopulation model **at the individual household level (32x32 grid)** could be approximated by varying parameters in the corresponding single patch model whose....”*

Results:

Line 201: It seems counter-intuitive that reducing the strength of density-dependent larval mortality would decrease the population growth rate. It would be helpful to the reader to clarify again the role

of Omega in your equations and that this isn't a linear decrease in density-dependence. This could also be mentioned in the methods (line 338).

This result arises because $\Omega < 1$ gives higher larval mortality than $\Omega = 1$ when the larval population is below the carrying capacity during the early stages of population growth following seeding. In turn, this results in slower population growth following seeding, and therefore produces a population growth curve more similar to that observed from the metapopulation model. In addition, we found that reducing carrying capacity alone allowed us to approximate mean equilibrium total population size (Figure 7a,c). Thus, combining these two elements (reduced K and $\Omega < 1$) allowed us to reduce both the speed of population growth and equilibrium population size, thereby providing the best approximation to the metapopulation model results.

Additional text has been added to the Methods and Results sections clarifying the role of Ω .

Methods Section:

*"... $M_{ij}(t)$ denotes the net migration of adult mosquitoes into (i, j) at time t, Ω describes the **power on density dependence (with $\Omega = 1$ describing linear density-dependence)**, b denotes the oviposition rate,...." (page 18)*

*"We explored if the invasion dynamics observed **for a homogeneous landscape** under the metapopulation model **at the individual household level (32x32 grid)** could be approximated by varying parameters in the corresponding single patch model whose values are not constrained by directly observed ecological processes - namely mean larval carrying capacity (K), and **the power on density dependent larval mortality (Ω)**. Hence, by varying , we allow for **non-linear changes in the larval mortality rate as the larval population grows.**" (page 26)*

Results Section:

*"Instead, allowing both K and Ω to vary resulted in a better approximation of the growth rate of the population (Figures 7a,b, Supplementary Fig. 2a-d, Supplementary Fig. 3a-d), with reduced K and Ω giving the best fitting combination of values for fragmented landscapes where individual patches have a low carrying capacity (Supplementary Fig. 4). This is because $\Omega < 1$ (**giving sub-linear density-dependence**) and reduced K results in higher larval mortality and thus slower population growth when larval density is low **during the early stages of population growth following seeding**, compared with single patch models where $\Omega = 1$ and K is larger. This in turn allowed us to better replicate the full temporal curve of population growth observed under the metapopulation model." (page 12)*

Figure 3d: legend text is overlapping, making number unreadable.

We thank the reviewer for noticing this. This has now been corrected (page 6).

Figure 3: Impact of Fragmentation when Seeding a Population. Population dynamics observed when a single patch in a landscape with temporal variation in carrying capacity is seeded with an adult mosquito population, and is modelled as a single patch (dashed lines) or as a metapopulation (with a dispersal rate of 0.08 per day) on a 32x32 grid (solid lines). We consider the dynamics for landscapes with a mean equilibrium larval population of 6 or 12 larvae per patch (L^*), seeded with 6 or 12 adult mosquitoes respectively, and where the amplitude and phase of seasonal variation in carrying capacity are 0.7 and 0.5 respectively. The results are compared with respect to (a) the mean proportion of models runs in which population persistence was achieved (final total adult population >1), (b) the mean proportion of patches occupied, (c) mean total adult mosquito population size, and (d) the standard deviation of total adult mosquito population size. **For each scenario, the mean and variance were calculated across 1000 realisations of the stochastic model.**

Figure 4: Please double-check that this figure legend is correct. It seems like a shorter dispersal distance would result in c/d, not a longer one, based on the text directly above.

We thank the reviewer for drawing this to our attention. There was an error in the figure legend – Figures 4c,d correspond to a mean dispersal length of 1 patch, not 5 patches as originally described. The figure legend has now been corrected (page 8).

Figure 4: Modelling at Different Levels of Spatial Granularity. Population dynamics observed when a landscape with temporal variation in carrying capacity and an equilibrium mean larval population of 6 larvae per patch (at the 32x32 level) is seeded with an adult population of 6 mosquitoes and modelled at different levels of spatial granularity. The dispersal rate was 0.08 per day. (a-b) correspond to dynamics observed when the mean dispersal length (at the 32x32 level) is **5 patches**, and (c-d) correspond to the dynamics observed when the mean dispersal length (at the 32x32 level) is **1 patch**. The maximum dispersal length (at the 32x32 level) is set to 32 patches. **For each scenario, the mean and variance were calculated across 1000 realisations of the stochastic model.**

Figure 7: The legend for this figure is slightly confusing. What is the difference between a/b and c/d? The legend suggests that c is the variance of the total population at each time step. Is the variance meant to be across simulations? I also am not sure that c is a depiction of variance. Please double-check these labels are correct.

In Figure 7 we provide an example of how the results obtained when approximating the invasion dynamics observed under the metapopulation model using an adjusted single patch model depend on which characteristics we sought to approximate. Figures 7a,b correspond to the scenario where we approximate the mean equilibrium total adult population size and the growth rate of the population. In Figures 7c,d, we approximate the variance of the population, in addition to mean equilibrium total adult population size and the growth rate of the population. The mean and variance were calculated across 1000 model simulations.

The figure legend has been amended to make this distinction clearer (page 12).

Figure 7: Single Patch Approximations-Example. Comparison of results obtained when we approximate the invasion dynamics observed when a homogeneous landscape comprised of 1024 patches, with an equilibrium larval population of 6 larvae per patch, is seeded with 6 adult mosquitoes. A mean dispersal length of 1 patch and a dispersal rate of 0.10 was used. (a,b): **This corresponds to the scenario where we approximate two quantities - the mean equilibrium total adult mosquito population and the growth rate of the population.** (c,d): **Here we approximate three quantities -We approximate the mean and variance of the equilibrium total adult mosquito population and the growth rate of the population. For each scenario, the mean and variance calculated across 1000 realisations of the stochastic model.**

In general, it would be wise to go through and ensure all of your figure legends are correct, as some seem to have panels mixed up in the legends.

We apologise for the mix up in some figures. All figure legends have been checked, and are now correct.

Line 198: Mislabeled figure reference.

This has now been corrected (page 12).

“Instead allowing both K and Ω to vary resulted in a better approximation of the growth rate of the population (Figures 7a,b, Supplementary Fig. 1a-d, Supplementary Fig. 2a-d),...”

Reviewer #2 (Remarks to the Author):

Dear Editor,

This paper presents an innovative mathematical model describing mosquito population dynamics considering spatial distribution of patches with different carrying capacity. Simulations using this model were carried out evidencing that spatial distribution of patches can modify significantly the final results on the mosquito population dynamics.

1. General critics over the presented model is that it relays on tens of parameters and particular distributions. It is not trivial how to obtain these data from experiments and thus how to use this

model as a quantitative study in the public health management. However, the model is perfectly valid for qualitative investigations.

We agree that the principal purpose of this work is qualitative investigation of the effect of metapopulation structure on mosquito dynamics. As such, we largely used parameter value estimates currently available in the literature. As noted by the reviewer, it is often very difficult to obtain empirical estimates of many important characteristics of mosquito populations (e.g. the mosquito dispersal rate). Hence while we believe our study provides important insights into these dynamics, we agree that theoretical modelling studies are unlikely to be able to fully capture every aspect of real-world dynamics.

Our main aim was to explore how allowing different levels of spatial heterogeneity in mixing among larval populations, and our choice of representation of spatial structure (if represented at all) when modelling mosquito populations influences our understanding of fine-scale mosquito population dynamics. Through explicitly modelling and comparing the dynamics of the same metapopulation at different levels of spatial granularity, we gained a deeper understanding of the key factors influencing fine-scale mosquito population dynamics, and of the importance of model choice in how to represent and account for the fragmented structure of larval populations. Thus, although model parameter values may vary according to species and location, we believe the overall conclusions drawn will hold more generally for models of mosquito population dynamics.

2. On the page 17 the basic mosquito reproduction number R is defined as a product of three terms. The two containing “gamma” represent the survivor probability and not mortality as stated in the text. The equation itself is correct. Also, it is not clear how the term R appears in the system (1)-(5).

We thank the reviewer for drawing our attention to this. The text on page 19 has now been corrected as advised above.

Here we use our equation for R_M to assign the value of parameter b in equations (1)-(5). This has now been clarified in the Methods section (page 19).

*“Here $\frac{b}{\mu_A}$ describes the average number of eggs laid by a female over the course of her lifetime, while the terms $\frac{\gamma_E}{\gamma_E + \mu_E}$ and $\frac{\gamma_L}{\gamma_L + \mu_L}$ describe the probability of survival during the egg and larval stages respectively when determining the average number of females produced. **Here, the value of b in equations (1)-(5) is assigned so that R_M remains fixed.**”*

3. Please pay attention to the punctuation in the equations (15), (16) and (17) on page 18.

This has now been corrected.

4. Mathematical models dealing with spatial dependence of carrying capacity and the impact on the mosquito population dynamics were addressed in other works. The authors may be interested in comparing the presented methodology with one proposed in Parasites & Vectors 2018, 11:245 (<https://doi.org/10.1186/s13071-018-2829-1>).

We thank the reviewer for highlighting this study. While there are similarities in the models used, the model details and fundamental motivations of that paper and ours differ. The Yamashita paper uses a deterministic diffusion model to explore the effect of city landscapes and airflow (wind) on mosquito dispersion. Our paper uses a stochastic model (modelling mosquito populations as integer

numbers of individuals, rather than continuous densities) to examine the effect of spatial heterogeneity on the persistence and growth dynamics of mosquito populations.

The following text has been added to the Discussion (page 16)

*“However, the rate at which mosquito larvae develop is temperature dependent^{19,57} and older adult mosquitoes may experience increased mortality^{58,59}. **A further limitation is that we have only studied random landscapes here. Work is needed to parameterise spatially explicit models of mosquito dynamics against data from real landscapes, such as cities (e.g. Yamashita et. al⁶⁰ who deterministically modelled the influence of urban landscapes and wind on mosquito dispersion).** Last, although appropriate values for model parameters were sourced where possible from the existing literature, the model presented was not explicitly fitted to entomological data.”*

REVIEWERS' COMMENTS:

Reviewer #1 (Remarks to the Author):

The authors have addressed all of my prior concerns and comments. I think this is a well-thought out, interesting study that will help improve spatial models of mosquito-borne diseases.

Reviewer #2 (Remarks to the Author):

The authors answered the questions satisfactorily and made all the proposed suggestions and corrections. In my opinion the study is new and interesting for the community.